# COVID-19 mRNA Vaccines: The Molecular Basis of Some Adverse Events

**DOI:** 10.3390/vaccines11040747

**Published:** 2023-03-28

**Authors:** Girolamo Giannotta, Antonio Murrone, Nicola Giannotta

**Affiliations:** 1Azienda Sanitaria Provinciale Vibo Valentia, 89900 Vibo Valentia, Italy; 2Oncologia Territoriale, Hospice Cure Palliative ASUFC, 33030 Udine, Italy; antonio.murrone@asufc.sanita.fvg.it; 3Medical and Surgery Sciences, Faculty of Medicine, Magna Græcia University, 88100 Catanzaro, Italy; nicola.giannotta@studenti.unicz.it

**Keywords:** COVID-19 mRNA vaccines, myo-pericarditis and COVID-19 mRNA vaccines, multisystem inflammatory syndrome and COVID-19 mRNA vaccines, arrhythmias and COVID-19 mRNA vaccines, pathogenesis of myocarditis following COVID-19 mRNA vaccines, MIS-A, MIS-C, MIS-V, myocarditis, COVID-19 mRNA vaccine adverse events

## Abstract

Each injection of any known vaccine results in a strong expression of pro-inflammatory cytokines. This is the result of the innate immune system activation, without which no adaptive response to the injection of vaccines is possible. Unfortunately, the degree of inflammation produced by COVID-19 mRNA vaccines is variable, probably depending on genetic background and previous immune experiences, which through epigenetic modifications could have made the innate immune system of each individual tolerant or reactive to subsequent immune stimulations.We hypothesize that we can move from a limited pro-inflammatory condition to conditions of increasing expression of pro-inflammatory cytokines that can culminate in multisystem hyperinflammatory syndromes following COVID-19 mRNA vaccines (MIS-V). We have graphically represented this idea in a hypothetical inflammatory pyramid (IP) and we have correlated the time factor to the degree of inflammation produced after the injection of vaccines. Furthermore, we have placed the clinical manifestations within this hypothetical IP, correlating them to the degree of inflammation produced. Surprisingly, excluding the possible presence of an early MIS-V, the time factor and the complexity of clinical manifestations are correlated to the increasing degree of inflammation: symptoms, heart disease and syndromes (MIS-V).

## 1. Introduction

In the post-marketing period, some adverse events (AEs) were temporally associated with the injection of COVID-19 mRNA vaccines [1,2,3,4,5,6,7,8,9,10,11,12,13]. Starting from the evidence that up to now no studies have been published on the pathogenetic mechanisms that could determine cardiac AEs, we have conducted research focused on the molecular effects determined by the Spike protein that have allowed us to propose a unifying mechanism that includes at least four groups of AEs: 1, systemic symptoms; 2, arrhythmias; 3, myo-pericarditis; and 4, multisystem hyperinflammatory syndromes (MIS).

Considering the clinical relevance of cardiac AEs (arrhythmias, myo-pericarditis), in this study we have devoted ample space to a series of pathophysiological mechanisms that can be influenced by the administration of lipid nanoparticles (LNPs) containing mRNA encoding Spike protein.

Furthermore, we compared, on the basis of the currently available scientific literature data, the risks and the characteristics of cellular infiltrates in myo-pericarditis following COVID-19 mRNA vaccines and after SARS-CoV-2 infection. Finally, we attempted to verify if after infection the risk of myo-pericarditis was different than the risk following vaccination, especially in younger subjects.

## 2. Human Heart

In the human heart, there are several cell types: cardiomyocytes (CMs), cardiac fibroblasts (CFs), cardiac endothelial cells (CECs), macrophages, smooth muscle cells, pericytes and other cells [14,15,16,17], but CMs are about one third of the entire cell population [17,18,19,20,21]. In CMs, there is an axis toll-like receptor 4 (TLR4)/nuclear factor kappa B (NF-kB) which is important in cardiac inflammation [22]. TLR4 activation triggers an inflammatory immune response via the TLR4/MyD88/NF-κB signaling pathway [23]. SARSCoV-2 Spike protein S1 subunit (CoV-2-S1) induces high levels of NF-κB activation [24].

CFs are the primary cell type responsible for the deposition of the extracellular matrix (ECM) in the heart, providing support to the contracting myocardium and contributing to a myriad of physiological signaling processes [25]. CFs contribute to both the electrical and structural remodeling of the heart, which ultimately leads to decreased cardiac function, heart failure and arrhythmias [26,27,28,29]. Myocardial interstitium is a complex and dynamic microenvironment [30], and ECM synthesis contributes to myocardial fibrosis [31]. When the ECM expands, the extracellular space expands and signals a process of myocardial fibrosis which can be detected with cardiac magnetic resonance (CMR) imaging, where an excess of gadolinium is retained and deposited in the extracellular space [32], producing the phenomenon of late gadolinium enhancement (LGE).

CECs form a barrier to the movement of fluids and molecules [33] and during inflammation the junctional proteins of the paracellular pathway are modified with a subsequent interruption of the integrity of this barrier [34]. CECs organize the recruitment of immune cells and regulate leukocyte extravasation at places of inflammation by the inducible expression of adhesion molecules, and maintain appropriate hemostasis or coagulation.

Several cardiac cells express angiotensin-converting enzyme 2 (ACE2) [35], but pericytes have the highest concentration of ACE2 and their injury may result in microvascular dysfunction [36], which facilitates the transit of neutrophils and macrophages in a pro-inflammatory environment [36].

## 3. Systemic Symptoms

In our representation of the hypothetical IP, at the first level we have placed the systemic symptoms that arise after the injection of the vaccine (muscle aches, chills, asthenia, fever, headache, pain at the injection site), symptoms generated by the strong production of pro-inflammatory cytokines, species interleukin (IL)-1 IL-1β, IL-6 and tumor necrosis factor (TNF)-α. In our previous studies, we have described the mechanism of action of common infant vaccines and found that there is no adaptive immune response to vaccine injection if the innate immune system is not activated, which also leads to the strong expression and secretion of various pro-inflammatory cytokines, such as IL-1β, IL-6 and TNF-α [37,38].

## 4. Inflammatory Cardiac Channelopathies and Arrhythmias

The immune system can promote cardiac arrhythmias by means of autoantibodies and/or inflammatory cytokines that directly affect the function of specific ion channels on the surface of CMs [39,40,41,42].

## 5. The Cardiac Action Potential (AP)

Normally, a cardiac muscle cell in a resting state has a negative electric charge of about 90 mV on the inner side of its cell membrane, compared to the surrounding medium. When an electrical stimulus excites a ventricular cell, the cell membrane rapidly depolarizes and the potential difference now becomes positive inside with a value of about 20 mV (positive overshoot). Now, we are at the 0 phase of action potential (AP). This is followed by phase 1 which consists of a short phase of partial repolarization that is followed by phase 2, which is that of the plateau that signals a period of intense and sustained depolarization of the membrane. The repolarization process starts at stage 3 and it is a slower process. The interval from the end of repolarization until the beginning of the next AP is designated as phase 4 [43].

## 6. The Ionic Basis of AP

The changes in the permeability of the cellular membrane are associated with the various phases of the AP and essentially involve three ions: Na^+^, K^+^ and Ca^2+^. These permeability changes alter the rate of ionic movements across the membrane [43]. Apart from the movement of sodium of the 0 phase of AP, the movements of potassium and calcium can be influenced by pro-inflammatory cytokines which act precisely on certain ion channels. In Table 1, we present a series of characteristics of some channels of the cardiac muscle fibrocells, relating them to the specific AP phase and we indicate the documented interferences that pro-inflammatory cytokines produce on ion channels and currents.

They produce three outward currents of K^+^: one in AP 1 (*I*_to_), and two in AP 3 (*I*_Kr_, *I*_Ks_). Conversely, Kir 2.1/2.3 channels produce an inward current in AP 4 (*I*_K1_). Pro-inflammatory cytokines interfere with the physiological function of all the currents represented here, excluding the inward current *I*_K1_.

Several studies have demonstrated the importance of voltage-gated potassium channels in the genesis of cardiac arrhythmias [44,45,46,47,48]. Loss-of-function mutations in the K_v_ 7.1 channel, lead to long-QT syndrome type 1 (LQTS1), which is the most frequent type of the long-QT syndromes [49]. The link to physiology is exemplary: patients with LQTS1 most frequently have arrhythmic events during exercise, where the sympathetic drive increases heart rate but fails to reduce the repolarization time, because the Kv 7.1 channel is dysfunctional [50]. Furthermore, TNF-α causes a functional inhibition of the K_v_ 7.1 channel which reduces the current I_k_ [51]. KV11.1 is essential for normal cardiac electrical activity and rhythm [52]. IL-6 produces an inhibition on the rapid delayed rectifier current *I*_kr_. The consequent prolongation of the duration of AP results in QT interval prolongation [53]. Finally, Kir 2.1/2.3 channels also determine the effects on AP [54,55,56].

Voltage-activated Ca^2+^ channels represent the major route of Ca^2+^ entry into CMs in response to depolarizations of the cellular membrane potential [57]. These channels are important in the AP plateau phase and are important for the electrical and mechanical properties of the heart [57,58,59,60,61,62,63].

## 7. Pro-Inflammatory Cytokines

Inflammatory factors can cause cardiac K^+^ channel dysfunction and arrhythmias in the setting of a structurally normal heart [64]. Emerging experimental evidence has shown that inflammatory responses, mainly via IL-1β, IL-6 and TNF-α, regulate CMs’ electrophysiological properties [65,66,67,68,69]. Pro-inflammatory cytokines can be arrhythmogenic and cytokines can promote the development of LQTS [40]. Some studies report in detail the effects of pro-inflammatory cytokines on cardiac ion currents [32,53,59,70,71,72].

Chiu et colleagues [73] published a study of 4928 students who were examined with serial ECGs performed before and after vaccination with the BNT162b2 COVID-19 vaccine. Of the students examined, 17.1% had at least one cardiac symptom after the second vaccine dose, mostly chest pain and palpitations. The incidence of cardiac adverse effects was reported to be as high as 1.5 per 10,000 persons after the second dose of the BNT162b2 COVID-19 vaccine in the young male population based on the reporting system. Through this mass ECG screening study, after the second dose of the BNT162b2 vaccine these authors found that the depolarization and repolarization parameters (QRS duration and QT interval) decreased significantly after the vaccine with increasing heart rate. Furthermore, the incidence of post-vaccine myocarditis and significant arrhythmia are 0.02% (2 per 10,000) and 0.08% (8 per 10,000).

## 8. Myo-Pericarditis

The mean monthly number of cases of myocarditis or myo-pericarditis during the pre-vaccine period was 16.9 vs. 27.3 during the vaccine period [74]. Myocarditis incidence after RNA vaccines is very rare (0.0035%) and has a very favorable clinical course [1]. However, not all cases are benign and critical or fatal cases have been reported [2,3,4,5]. There is an excess of cases with a substantial burden of both myocarditis and pericarditis in all ages [6]. An increased risk of myocarditis/pericarditis has been associated with the second dose of BNT162b2 and both doses of mRNA-1273 [7]. In one study, the highest risks were observed in males of 12 to 39 years [7], while Wong and colleagues [8] demonstrated that an increased risk of myocarditis or pericarditis was observed after the COVID-19 mRNA vaccination and was highest in men aged 18–25 years after a second dose of the vaccine.

### 8.1. Probable Pathogenesis of Myocarditis

Husby and Kober [75] argue that the disease mechanism is specific neither to the newly developed mRNA vaccines nor to exposure to the SARS-CoV-2 Spike protein. However, we try in this study to elaborate a pathogenetic hypothesis on the basis of extensive scientific documentation in support of our thesis.

Before proceeding, it is necessary to understand that the two vaccines have a partly different composition, both in the quantity of lipid nanoparticles (LNPs) and in the excipients. There is a difference in the amount of LNPs contained in the two vaccines: in the dose of Comirnaty (BNT162b2—Pfizer/Biontech), to be administered to subjects aged >12 years, there are 30 micrograms/dose [76]; while in the Moderna vaccine (mRNA-1273) there are 100 micrograms/dose [77]. As it is evident, the content of LNPs is three times higher in the mRNA-1273 vaccine, compared to BNT162b2, and could probably affect the timeline of the onset of these adverse events (AEs). In addition, the vaccination schedule provides an interval between the first and second dose of 21 days for the BNT162b2 vaccine [76], while this interval is 28 days if the mRNA-1273 vaccine is used [77].

#### 8.1.1. One or Two Shots?

In several studies, myocarditis/pericarditis are more frequent after an injection of the second dose of the BNT162b2vaccine, while myocarditis/pericarditis can occur both after the first and second injection of the mRNA-1273 vaccine. Considering the time interval between the two vaccine doses, one would think that it takes less than 28 days to produce these AEs (about 23–27 days for the BNT162b2 vaccine).

We must develop our hypotheses starting from the fact that there are two different formulations of COVID-19 mRNA vaccines and two different vaccination schedules. Furthermore, our immune system must produce dose-dependent inflammatory responses after the administration of two different doses of LNPs (30 micrograms/dose vs. 100 micrograms/dose). In addition, the immune response is complex and affects the entire LNP.

It was demonstrated that if the antigenic stimulus persists for over a week at low levels, it will cause chronic low-level stimulation of the T cells, keeping them in a partially activated state and leading to their accumulation over time [78,79,80].

Li and colleagues [81] conducted a study in a Balb/c mouse model and found that an intravenous injection of a COVID-19 mRNA vaccine rapidly resulted in multifocal myo-pericarditis, while an intramuscular injection produced signs 7 days after injection. (myocardial edema, and occasional foci of cardiomyocyte degeneration). However, these histopathological changes worsened after the second dose. Since the injected vaccine was BNT162b2, this study may indicate that two shots are needed with this vaccine to produce myo-pericarditis, as occurs predominantly in humans.

#### 8.1.2. Inflammatory Infiltration of Myocardium

Verma and colleagues [4] described two cases (one fatal) with multifocal myocardial damage (biopsy and autopsy) associated with mixed inflammatory infiltration. The surviving subject presented at the endomyocardial biopsy (EMB) an infiltration of T cells, macrophages, eosinophils, B cells and plasma cells. Conversely, the fatal case presented at the autopsy an infiltrate that was similar but did not contain plasma cells. Oka and colleagues [82] described a case of fulminant myocarditis after the second dose of COVID-19 mRNA vaccination. The inflammatory infiltrate was identical to the non-fatal case described by Verma and colleagues [4]. Kazama and colleagues [83] described the case of a woman with fulminant myocarditis following the second dose of mRNA-1273. Again, an EMB revealed lymphocytic infiltration with predominant immunostaining for CD8 and CD68-positive cells (macrophages). Two other cases with acute myocarditis following COVID-19 mRNA vaccination, at EMB analysis, have elevated CD3 T cells and CD68 macrophages [84]. The activation of T lymphocytes and macrophages [85,86,87,88,89] is believed to play a fundamental role in myocardial inflammation [85], and cardiac macrophage populations are markedly perturbed by inflammation [86].

Since COVID-19 mRNA vaccine-related myocarditis develops rapidly in 3 to 4 days after vaccination, innate immunity more likely contributes to the pathogenesis of vaccine-related myo-pericarditis than adaptive immunity [74]. It is useful to remember that any vaccine determines a strong expression of pro-inflammatory cytokines (including IL-18) by DCs, which is associated with an evident activation of NFκB [37,38]. In an inflammatory microenvironment, caspase-1 is regulated by NF-κB [90], and this enzyme facilitates the conversion of pro-IL-18 in IL-18. Patients with myo-pericarditis following a COVID-19 vaccination had excessive Th1-type immune responses over vaccine-induced immune activation. Diffuse myocardial macrophages infiltration in the patient biopsy sample suggest an increased level of IL-18 produced by monocytes and macrophages in the heart with COVID-19 vaccine-related myo-pericarditis [91].

#### 8.1.3. Who Opens the Gate

Now it is necessary to understand how the injection of the vaccine determines a series of immune events that can open a passage in the endothelial line of the heart vessels through which inflammatory cells will then infiltrate and can cause damage to the myocardium. We have identified at least four main actors: 1, DCs-derived exosomes; 2, pro-inflammatory cytokines; 3, adhesion molecules; and 4, Spike protein. The framework is then integrated by the activation of the TLRs and the TLR4/NFκB axis.

#### 8.1.4. Exosomes

Exosomes are cell-derived small extracellular membrane vesicles, 50–100 nm in diameter, whichare actively secreted and released both in physiological and pathological conditions. Exosomes contain and transport multiple types of biological macromolecules that maintain their whole activity when delivered to target cells [92]. This bioactive cargo includes nucleic acids, lipids and soluble or membrane-bound proteins [93]. Exosomes, produced by mature DCs, can produce inflammation in endothelial cells through their TNF-α content in their membrane, via transcription factor NF-κB, which also induces the transcription of adhesion molecules such as vascular cell adhesion protein 1 (VCAM-1), intercellular adhesion molecule 1 (ICAM-1) and E-selectin in endothelial cells (ECs) [94]. Furthermore, DCs-derived exosomes contain major histocompatibility complex (MHC) Class I and II molecules and T cell co-stimulatory molecules, which determine a direct antigen presentation and CD4 and CD8 T cell activation [34,95]. There is important crosstalk between CMs and CECs that is ensured by means of the cardiomyocytes-derived exosomes that are up-taken by CECs. ECs can also actively collaborate with the underlying CMs and modulate cardiac function, both under physiological conditions and under pro-inflammatory conditions [34,96,97,98,99,100,101,102].

In summary, after an injection of COVID-19 mRNA vaccines, DCs-derived exosomes may contain a variable amount of what is found in their cytosol: Spike protein-encoding mRNA, Spike protein, Spike protein peptides, pro-inflammatory cytokines, MHC Class I and II molecules and T cell co-stimulatory molecules that make the particle capable of initiating and maintaining an inflammatory process in the target tissue.

#### 8.1.5. Spike Protein Induces Endothelial Cells (ECs) Dysfunction

Spike protein of SARS-CoV-2 alone activates ECs inflammatory phenotype in a manner dependent on integrin ⍺5β1 signaling and induces the nuclear translocation of NF-κB and subsequent expression of leukocyte adhesion molecules (VCAM-1 and ICAM-1), coagulation factors, pro-inflammatory cytokines (TNF-α, IL-1β and IL-6), and ACE2. Furthermore, in vivo, intravenous injections of the protein Spike increases the expression of ICAM-1, VCAM-1, CD45, TNF-α, IL-1β and IL-6 in the lung, liver, kidney and eye [103].

Therefore, it emerges with extreme clarity that the protein Spike alone is able to induce the secretion of pro-inflammatory cytokines (via NF-κB) and adhesion molecules (ICAM-1 and VCAM-1) in ECs.

##### Spike Protein and Cardiomyocytes (CM)

CoV-2-S1 interacts with the extracellular leucine rich repeats-containing domain of TLR4 and activates NF-κB [104]. TLR4 initiates the expression of several pro-inflammatory genes, cell surface molecules and chemokines through the MyD88-dependent pathway, which exacerbates the damage to the myocardium [22]. The circulating CoV-2-S1 is a TLR4-recognizable alarmin that may harm the CMs by triggering their innate immune responses [104]. In CMs, there is an axis TLR4/NF-kB [22] and unmitigated TLR4 activation may lead to an increased risk for cardiac inflammation [105]. Thus, the TLR4/NF-kB axis in CMs can also cause cardiac inflammation and myocardial damage, and the Spike protein alone is capable of activating this axis in CMs.

Spike protein alone can easily reach the myocardium through these routes: 1, via DCs derived exosomes; 2, via ECs that have ACE2 receptors, and, after uptake, they can produce exosomes to be exported to CMs; and 3, via transmigration through the endothelium.

Baumeier and colleagues [106] studied 15 cases of myocarditis after a COVID-19 mRNA vaccine using EMB and immunohistochemical analysis. In nine of these patients, the Spike protein was found in CMs. Furthermore, CoV-2-S1 activates TLR4 signaling to induce pro-inflammatory responses in murine and human macrophages [107]. Finally, SARS-CoV-2-induced myocarditis and multiple-organ injury may be due to TLR4 activation, aberrant TLR4 signaling and hyperinflammation in COVID-19 patients [108].

#### 8.1.6. Young Males: The Favorite Target

Adolescence is accompanied by increased exposure to stressors [109,110] and it is a time of many psychosocial and physiological changes [111]. Varma and colleagues [112] showed that during the COVID-19 pandemic, younger subjects (18–35 years) developed a significantly higher stress level (*p* < 0.001) than older subjects. Furthermore, the stress response system (SRS) changes in adolescents. Acute stress prepares the body for astressful situation [113]. Briefly, theautonomic nervous system (ANS) responds to stress within milliseconds to minutes and the HPA axis responds over minutes to hours following stressor onset [114,115]. The stress response to physical and/or psychological stressors is initially produced by the ANS which introduces the catecholamines epinephrine (E) and norepinephrine (NE) into circulation [116].

Acute stress induces an inflammatory response and raises the circulatory levels of inflammatory cytokines [117], such as IL-6 and TNF-α, and also results in platelet activation and endothelial stimulation [118]. Mental stress induces prolonged endothelial dysfunction [119]. Adrenaline released during acute stress greatly increases both the inotropic and chronotropic effects in the heart via the β_2_ receptors [120]. Β_2_-receptor activation has effects on a calcium current (*I_CaL_*) and a potassium current (*I*_Ks_) which exert opposite effects on the AP. When *I*_Ks_ is dysfunctional, as in LQT1, the resulting unbalanced *I_CaL_* effect causes excess AP prolongation [121].

Overall, acute stress creates a pro-inflammatory and pro-arrhythmic condition that can worsen if eating disorders (typical of adolescents) coexist. For example, repeated vomiting leads to a loss of potassium and consequent hypokalemia, resulting in effects on other potassium ion currents, delays repolarization and promotes LQTS by suppressing K^+^ currents such as *I*_Kr_ and the background inward rectifier I_K1_ [122]. Chronic stress is more harmful, and in an animal under chronic stress, a new stressful stimulus determines a strong response from the ANS with a high secretion of catecholamines [123,124].

Stress reactions in Italian adolescents in response to the COVID-19 pandemic during the peak of the outbreak seem to be considerable [125]. Furthermore, we must not forget that young people practice sports of varying intensity. Combined stress (psychological and physical) can exacerbate cardiovascular responses to stress [126].

#### 8.1.7. Other Pathological Mechanisms Triggered by the Spike Protein

ACE2 is expressed throughout the cardiovascular system [127]. When the Spike protein binds to the ACE2 receptor, the typical enzymatic function (carboxypeptidase) is replaced by a receptor function which activates intracellular signaling pathways [128,129,130]. As a result, normal ACE2 activity in the renin–angiotensin–aldosterone system (RAAS) is lost and excess angiotensin II occurs. Furthermore, healthy young subjectswitha relative deficiency of some angiotensinases do not counterbalance ACE2 internalization, downregulation ormalfunction due to free-floating Spike proteins interactions, resulting in an increased risk of Ang II accumulation and adverse reactions. Thus, the adverse reactions to COVID-19 vaccinations associated with Ang II accumulation are generally more common in younger and healthy subjects [131].

##### Renin–Angiotensin–Aldosterone System (RAAS)

Renin, angiotensinand aldosterone represent the core of a complex hormonal axis, referred to as RAAS, which contributes to blood pressure control, sodium reabsorption, inflammation, and fibrosis [132]. The renin enzyme degrades angiotensinogen producing angiotensin I (Ang I). ACE catalyzes the transformation of Ang I to Ang II. Ang II, the primary physiological product of the RAAS system, is a potent vasoconstrictor [133]. ACE2 converts Ang II into Ang-(1–7) and activates the protective axis AT_2_R. Conversely, if the action of ACE2 is reduced, the AT_1_R axis is enhanced, which is pro-inflammatory, pro-apoptosis, and pro-fibrosis. The Ang II/AT_1_R axis is also involved in oxidative stress that stimulates endothelial dysfunction, inflammation of the vessels, and thrombosis [134,135,136]. Furthermore, the upregulated expression of AT_1_R is linked with arrhythmias [137] and cardiac remodeling [138].

It is interesting to remember that the Ang II/AT_1_R axis mediates a cascade of signals that induce transcriptional regulatory molecules NF-κB and AP-1/c-Fos via MAPK activation, and increased IL-6 release [139]. In addition, angiotensin II also promotes the expression and production of adhesive and pro-inflammatory molecules (VCAM-1, ICAM-1, monocyte chemoattractant protein-1, macrophage inflammatory protein-1α, and IL-8) on the endothelial and vascular smooth muscle cells [140,141,142,143,144,145]. The functional deficit of the Ang II/AT_2_R/Mas receptors axis does not produce the normal beneficial effects during stressful situations, lacking the ability to effectively modify sympathetic nervous activity during stressful conditions [145].

After vaccination, the free-floating Spike proteins released by the cells targeted by vaccines may interact with the ACE2 of other cells, thereby promoting ACE2 internalization and degradation (downregulation) [131,146]. Decreased ACE2 levels may lead to Ang II upregulation and over-activity of the classical RAAS axis (ACE, angiotensin II, and AT_1_R) [147].

##### Toxicity of Lipid Nanoparticles (LNPs)

LNPs are composed of cholesterol, a helper lipid, a polyethylene glycol (PEG) lipid and an ionizable amine-containing lipid [148]. Some cationic/ionizable lipids contained in the LNPs of COVID-19 mRNA vaccines pose toxicity problems [149]. Overall, LNPs exhibit a powerful pro-inflammatory action [150,151]. Small amounts of double-stranded RNA (dsRNA) can occasionally be packaged within mRNA vaccines [152]. LNPs evoke a strong pro-inflammatory response by activating the TLR pathways [151,152,153,154,155], and the inflammatory milieu induced by the LNPs could be partially responsible for the reported side effects of COVID-19 mRNA vaccines in humans [156]. Furthermore, the Spike protein present on plasma membranes could expose these cells to attack by anti-Spike antibodies, generating an antibody-dependent cellular cytotoxicity (ADCC) [157].

The BNT162b2 vaccine contains two novel LNP-excipients: ALC-0315 (aminolipid) and a polyethylene glycol lipid (PEG) ALC-0159 [158]. Saadati and colleagues [159] have recently described a new and clean way to produce ALC-0315, which contrary to the publicdomain route, does not use dangerous reagents, such as hexavalent chromium Cr (VI) which is cardiotoxic [160,161].

#### 8.1.8. TCD8, TCD3/CD45R0,and Macrophages CD68 in Myo-Pericarditis

Human T-cell differentiation should be delineated using a minimum set of canonical markers, i.e., CD45R0 (or CD45RA), CCR7, CD28, and CD95 [162]. CD45RA^−^CD8^+^ memory T cells expressing CD27 produce both IL-2 and IFN-γ but lack immediate killing activity (T memory cells); while effector T cells (CD45RA^−^CD8^+^CD27^−^) produce mostly IFN-γ and TNF-α, but not IL-2 and are capable of immediate cytotoxicity ex vivo [163]. CD45R0 is preferentially expressed on memory cells [164,165].

Under normal conditions, T lymphocytes also exercise immune surveillance on the heart with a continuous trafficking of T cells through the blood, lymphoid organs, and the heart [166]. The human CD8 T cell subset contains two distinct and separate entities: memory-and effector-type T cells, and the degree of systemic inflammation produced by vaccination affects the phenotype of secondary memory CD8 T cells [167]. Upon the second encounter with the cognate antigen, memory T cells are ready to proliferate and perform cytotoxic functions [168].

### 8.2. Diagnostic Items

The definition of myocarditis has been more recently enumerated by the ESC Working Group on Myocardial and Pericardial Diseases [169]. In patients with myocarditis, the main inflammatory populations consisted of macrophages and T cells [170,171,172].

Diagnostic tools include clinical signs and symptoms (difficulty breathing, chest pain), auscultation, ECG, echocardiography, cardiac MRI (to detect late gadolinium enhancement or LGE), and EMB, when indicated. Among the laboratory markers that are useful: creatine kinase MB, troponin I, and PCR [173]. For all the details, see the position statements [169,173].

Autoimmune forms of myocarditis excluded [174,175], immunohistochemical examinations [176,177] uniformly confirm that the inflammatory infiltrate is composed of activated CD3 T lymphocytes and CD68 macrophages (the human heart contains distinct macrophage subsets) [178]. Conversely, in the autoimmune forms, cardiac antigen-specific CD8 T lymphocytes could also be produced due to the molecular mimicry between Spike peptide and myocardial antigens [179].

### 8.3. Trans-Endothelial Migration towards Heart Tissue

Leukocyte homing and recruitment require the adhesion of leukocytes to the endothelial lining of postcapillary venules, a process that involves molecules on the surfaces of both the leukocytes and endothelial cells [180,181,182].

### 8.4. Immune Black Hole

We presume to know a series of events following the injection of the vaccine and to predict reasonably enough what happens after the second injection; while we do not know how the immune system behaves in the period of time between the injection of the vaccine and the clinical onset of myo-pericarditis. We know that a few hours after the injection of the vaccine, there is a strong production of pro-inflammatory cytokines and the synthesis of adhesion molecules is enhanced. All this makes it easy for T lymphocytes and macrophages to migrate towards the myocardium. Memory CD8 T cells, upon reinfection and antigenic stimulation, have the capacity to rapidly proliferate and differentiate into secondary effector CD8 T cells [183,184].

### 8.5. Experimental Myocarditis

There are two different study models of myocarditis: infectious and non-infectious. In non-infectious models, myocarditis is typically triggered by an autoimmune response against heart-specific antigens [185]. α-isoform of cardiac myosin heavy chain (α-MyHC) is not expressed in cells implicated in T cell tolerance. This results in the undisturbed development of α-MyHC-specific T cells in human [186] due to the molecular mimicry between Spike protein and α-MyHC. In fact, antibodies directed to SARS-CoV-2 Spike glycoproteins might cross-react with structurally similar human protein sequences, including myocardial α-MyHC [185].

In summary, these two experimental models do not add contributions to our understanding because the cases following the injection of vaccines are not of an infectious nature and there are few cases of autoimmune etiology.

## 9. Multisystem Inflammatory Syndromes (MIS)

Multisystem Inflammatory Syndrome in children (MIS-C) is a new pediatric illness that is a late complication of SARS-CoV-2 infection. Myocardial dysfunction with or without mild coronary artery dilation can occur in MIS-C [187]. MIS-C is characterized by a fever, systemic inflammation, and multisystem organ involvement [188,189,190] with particular interest in the gastrointestinal, cardiovascular, and neurological systems, associated with elevated markers of inflammation and altered coagulation [191]. MIS in adults (MIS-A) is a serious hyperinflammatory condition that presents approximately 4 weeks after the onset of acute COVID-19 with extrapulmonary multiorgan dysfunction [192].

MIS following COVID-19 vaccination (MIS-V) remains a rare event [193], but it is a serious adverse event. There have been 12 cases identified of hyper-inflammatory syndrome following COVID-19 mRNA vaccines (now MIS-V) in 12–17-year-old children in France, with multisystemic involvement [194], and two pediatric cases with neurological involvement in Italy [195]. Two other cases have recently been described and both children presented with MIS-V within 4 and 5 weeks of receiving their first and only dose of the BNT162b2 mRNA vaccine [196]. There are other MIS-V case reports and case series.

It follows that the cases of hyperinflammatory syndromes triggered by the injection of the COVID-19 mRNA vaccines must now be indicated by the acronym MIS-V.

## 10. Discussion

The injection of COVID-19 mRNA vaccines results in a strong expression and secretion of pro-inflammatory cytokines associated with a wide and variable cellular activation, both immune and vascular. In relation to the degree of inflammation produced by each subject, depending on its genetic status and the acquired condition of epigenetic modification of the innate immune system, systemic symptoms, heart disease, and hyperinflammatory syndromes can be produced as AEs.

In Table 2, some effects determined by the injection of COVID-19 mRNA vaccines are listed.

**Table 2 vaccines-11-00747-t002:** Summary of some effects determined by the injection of COVID-19 mRNA vaccines.

Potential Effects after COVID-19 mRNA Vaccination.	Ref.
Endothelial dysfunctions produced by the Spike protein.	[103]
After vaccination, angiotensin II accumulates.	[131]
The binding of Spike protein to ACE2 receptors promotes their downregulation.	[131,146]
Decreased ACE2 levels may lead to Ang II upregulation and over-activity of the classical RAAS axis (ACE, angiotensin II, and AT_1_R) resulting in the production of all the effects already described in this study [127,128,129,130,132,133,134,135,136,137,138,139,140,141,142,143,144,145].	[147]
Excessive Th1-type immune responses.	[197]
Persistence of the Spike protein in circulation for a prolonged period of time.	[198]
Prolonged immune and inflammatory response against the Spike protein.	[198,199]
Strong pro-inflammatory activity of LNPs.	[149,150,151,152,153,154,155,156]
Spike protein alone can easily reach the myocardium.	[104,106]
Spike protein was found in cardiomyocytes (CMs).	[106]
Different levels of expression of pro-inflammatory cytokines over time.	[199,200]
Circulating CoV-2-S1 is a TLR4-recognizable alarmin that may harm the CMs by triggering their innate immune responses.	[104]
TLR4 initiates the expression of several pro-inflammatory genes, cell surface molecules, and chemokines through the MyD88-dependent pathway, which exacerbates the damage to myocardium.	[22]
Activation of TLR4 and the TLR4/NFκB axis in cardiomyocytes by the Spike protein.	[24]
Unmitigated TLR4 activation may lead to increased risk for cardiac inflammation.	[105]
CoV-2-S1 activates TLR4 signaling to induce pro-inflammatory responses in murine and human macrophages.	[107]
Diffuse myocardial macrophages infiltration in the patient biopsy sample suggest an increased level of IL-18 produced by monocytes and macrophages in the heart with COVID-19 vaccine-related myo-pericarditis.	[91]
In an inflammatory microenvironment, caspase-1 is regulated by NF-κB, and this enzyme facilitates the conversion of pro-IL-18 in IL-18.	[90]
Lymphocytic infiltration with predominant immunostaining for CD8 and CD68-positive cells (macrophages) is present in myocarditis following COVID-19 mRNA vaccines.	[83]
Vaccinated mice showed signs of myocarditis 2 days after injection of the second dose of BNT162b2 vaccine.	[81]
Free Spike antigen was detected in the blood of adolescents and young adults who developed myocarditis following COVID-19 mRNA vaccine.	[201]
SARS-CoV-2 Spike protein induces inflammation via TLR2-dependent activation of the NF-κB pathway.	[202]

Different levels of expression of pro-inflammatory cytokines over time, after a COVID-19 mRNA vaccination [199], the persistence of the Spike protein in circulation for a prolonged period of time [198], the prolonged immune and inflammatory response against the Spike protein [198,199], the strong pro-inflammatory activity of LNP [149,150,151,152,153,154,155,156], the actions of the Spike protein on the Angiotensin II/AT_1_ axis [127,128,129,130,131,133,137,146,147], the activation of TLR4 and the TLR4/NFκB axis in cardiomyocytes by the Spike protein [24], and the endothelial dysfunctions produced by the Spike protein [103] all together represent a series of subsets that can contribute with variable expression, especially to the pathogenesis of myocarditis and multisystem syndromes.

Biochemical studies revealed that Spike protein triggers inflammation via activation of the NF-κB pathway and the induction of pro-inflammatory cytokines, such as IL-6, TNF-α, and IL-1β [199]. Furthermore, the expression of cytokines and chemokines, in response to Spike protein, was dose-dependent and this agrees with the different timeline of myo-pericarditis following COVID-19 mRNA vaccines (onset after second dose of BNT162b2 vaccine or at first and second dose of mRNA-1273 vaccine). After the first dose of the BNT162b2 vaccine, the human organism produces systemic inflammation which is accompanied by the upregulation of TNF-α and IL-6 after the second dose [200].

Furthermore, the S1 subunit of the Spike protein produces an endothelial lesion that is amplified by simultaneous exposure to the inflammatory cytokine TNF-α and the male hormone dihydrotestosterone [203]. This condition of endothelial lesion, amplified by simultaneous exposure to TNF-α and androgens, may allow us to resolve some controversies. There is growing evidence that suggests that males have a higher risk of outcomes in the case of myocarditis [204], despite the fact that they are able to suppress the production of pro-inflammatory cytokines (IL-1β, IL-6 and TNF-α) and increasing the production of anti-inflammatory cytokines [205]. Since the effects of testosterone may be different under normal physiological conditions and in pathological states [206], in the presence of an endothelial lesion and/or myocarditis these effects may be different from the physiological conditions. Indeed, generally, androgens have been found to increase Th1 responses [197] and, in acute myocarditis, testosterone promotes the pro-inflammatory Th1 and/or Th17-type immune response [207] and increases the activity of the inflammasome and TLR4 signaling pathways [208].

This synergy of effects could explain why myo-pericarditis is more frequent in males, while the concomitant pro-inflammatory action of stress and its ability to induce endothelial dysfunction could explain why young males are more affected by myo-pericarditis.

Exosomes with Spike protein, Abs to SARS-CoV-2 Spike, T cells secreting IFN-γ, and TNF-α increased following the booster dose [198]. Miyashita and colleagues [199] investigated the correlation between pro-inflammatory cytokine levels in sera and AEs after the COVID-19 vaccination, and they found that systemic TNF-α levels were connected with the systemic scores after the second dose. This observation also supports the notion that pro-inflammatory cytokines are a cause of AEs after vaccination [209,210,211,212]. Furthermore, Miyashita and colleagues [199], in the same study, measured serum pro-inflammatory cytokine levels after vaccination. IL-6 levels one day after the first dose were elevated compared with the levels before vaccination, and the levels were further elevated after the second dose. Serum TNF-α levels did not increase after the first dose but increased significantly after the second dose. It follows that after the second dose of the vaccine, there are markedly increased concentrations of IL-6 and TNF-αin the serum, already only after the first day following the second vaccine dose. Finally, there would be a significant linear correlation (*p* < 0.05) between the levels of pro-inflammatory cytokine TNF-α and the degree of symptoms (systemic scores) occurring one day after the second dose of the BNT162b2 vaccine. For these authors, these data suggest that pro-inflammatory cytokines (IL-6 and TNF-α) were produced in response to the BNT162b2 vaccination, especially after the second dose. Murata and colleagues [213] published a study reporting that four subjects died after receiving a second dose of the COVID-19 vaccine, with no obvious cause identified at autopsy. RNA sequencing revealed that genes involved in neutrophil degranulation and cytokine signaling were upregulated in these cases, suggesting that immune dysregulation occurred after vaccination.

Flego and colleagues [214] demonstrated that the administration of the mRNA-based vaccine BNT162b2 determines, in some subjects, a rapid increase in the systemic concentration of a series of pro-inflammatory cytokines (including IL-1β, TNF-α and IL-18) within 3–10 days after the first injection and 10 days after the second dose. Thus, one month after the first dose, we have a second wave of pro-inflammatory cytokines expression which coincides with the timeline of the onset of myo-pericarditis. The result of the increase in the serum concentration of IL-18 is relevant, since myo-pericarditis following COVID-19 mRNA vaccination may be associated with increased IL-18-mediated immune responses and cardiotoxicity [91].

Furthermore, COVID-19 vaccines were associated with rhythm disorders (inflammatory cardiac channelopathies) [215], and vaccination fear, as an acute stress situation could lead to atrial arrhythmias [216]. Lazzerini and colleagues [39,40,41,42,51] have studied inflammatory cardiac channelopathies in the past and the role of pro-inflammatory cytokines in producing arrhythmias is now well established. Esposito and colleagues [217] believe that among the main mechanisms associated with the development of myocarditis after vaccination with COVID-19 mRNA vaccines, these elements could be considered: activation of natural killer lymphocytes and macrophages and a massive release of cytokines leading to massive damage to the heart tissue.

Acute myocarditis is an inflammatory myocardial disease, which can be complicated by adverse cardiac events, including sudden cardiac death and heart failure [218].

From a series of epidemiological studies [1,2,6,7,8], it emerges that there is an evident excess of myo-pericarditis in all ages, especially in young people who have been vaccinated with COVID-19 mRNA vaccines, compared to the pre-vaccination period.

In a nationwide cohort study [219], cases of myo-pericarditis after the second dose of heterologous vaccination (a first injection with a ChAdOx1 vaccine followed by mRNA vaccine injection) are almost equal to cases occurring after two doses of a homologous vaccination with mRNA vaccines (0.028% vs. 0.027%). Conversely, at the third dose of vaccine with the heterologous scheme, there are no cases of myo-pericarditis, while in the homologous vaccination with mRNA vaccines the percentage of myo-pericarditis is 0.014%.

Oster and colleagues [9] studied 1626 cases of myocarditis reported in a national passive reporting system. The rates of myocarditis cases were the highest after the second vaccination dose in males aged 12 to 24 years with the highest incidence in the age group being 16–17 years (105.9 per million doses of the BNT162b2 vaccine). In Israel, 136 cases of definite or probable myocarditis were recorded that had occurred in temporal proximity to the receipt of two doses of the BNT162b2 mRNA vaccine, a risk that was more than twice that among unvaccinated persons. This association was highest in young male recipients within the first week after the second dose. Approximately 1 case in every 6637 male recipients occurred over the age range of 16–19 years [10]. Buchan and colleagues [220] found that vaccine products and interdose intervals, in addition to age and sex, may be associated with the risk of myocarditis or pericarditis after receipt of these vaccines. Vaccine effectiveness may be higher with an interdose interval for mRNA vaccinations of 6 to 8 weeks compared with the 3to 4week interval [221]. It follows that the Intervals adopted between the first and second dose, on the one hand, reduce the effectiveness of the vaccine; while on the other hand they increase the risk of myo-pericarditis, with respect to greater intervals between the two doses.

Hence, the number of myo-pericarditis is important and undiagnosed cases could be numerically more important and clinically insidious, since an increase in extracellular matrix deposition could lead to electrical destabilization of the heart [222].

Husby and Kober [75] argue that the disease mechanism of myo-pericarditis is specific neither to the newly developed mRNA vaccines nor to exposure to the SARS-CoV-2 spike protein. However, we have found a number of elements that do not move in the same direction as indicated by Husby and Kober [75]. Furthermore, it does not appear from the published statistics that there are such an important number of cases of myo-pericarditis after the injection of a traditional vaccine [223,224]. Myocarditis associated with COVID-19 mRNA vaccines in adult males occurs with a significantly higher incidence than in the background population. [172]. The incidence of myo-pericarditis following COVID-19 mRNA vaccines varies from case to case, starting from the lowest data of Das and colleagues [12], which is 0.32/100,000, to arrive at the highest data of Nygaard and colleagues [13], which is equal to 5.74/100,000.

In a series of report cases of myocarditis following COVID-19 mRNA vaccination [4,82,83,84], studied with EMB, there is a mixed inflammatory infiltrate in which CD3 T lymphocytes and macrophages CD68 are always present. While CD4^+^ and CD8^+^ cell infiltration prevails in typical inflammatory myocarditis, CD68^+^ cell infiltration is prevalent in SARS-CoV-2-induced myocarditis [172]. The activation of T lymphocytes and macrophages is believed to play a fundamental role in myocardial inflammation [87].

Established that the activation of the innate immune system follows the injection of COVID-19 mRNA vaccines and that the migration of T lymphocytes and macrophages is a real fact in myocarditis, we will now examine the fundamental role of the Spike protein in modifying certain cell physiology events. After the injection of COVID-19 mRNA vaccines, the Spike protein is expressed in DCs at the level of the axillary lymph nodes ipsilateral to the injection site (deltoid muscle) [225]. These DCs produce exosomes that circulate in the blood for a long time [198]. Spike protein induces ECs dysfunction. Spike protein of SARS-CoV-2 alone activates ECs inflammatory phenotype and induced the nuclear translocation of NF-κB and subsequent expression of leukocyte adhesion molecules (VCAM-1 and ICAM-1), coagulation factors, pro-inflammatory cytokines (TNF-α, IL-1β and IL-6), and ACE2 [103]. CoV-2-S1 interacts with the extracellular leucine rich repeats-containing domain of TLR4 and activates NF-κB [104]. TLR4 initiates the expression of a number of pro-inflammatory genes, cell surface molecules, and chemokines through the MyD88-dependent pathway, which exacerbates the damage to the myocardium [22]. The circulating CoV-2-S1 is a TLR4-recognizable alarmin that may harm the CMs by triggering their innate immune responses [104]. In CMs, there is an axis TLR4/NF-κB, and unmitigated TLR4 activation may lead to increased risk for cardiac inflammation [22]. Thus, the TLR4/NF-kB axis in CMs can also cause cardiac inflammation and myocardial damage, and the Spike protein alone is capable of activating this axis in CMs. We have already indicated four different pathways that allow the Spike protein to reach the myocardium.

In summary, the Spike protein is not a mere spectator but the main protagonist in myocarditis. In fact, it causes endothelial dysfunction [103] and activates TLR4 and the TLR4/NFκB axis in CMs with often unhealthy consequences [24,104,105]. The concreteness of all these scientific works has been validated by clinical practice. Indeed, Baumeier and colleagues [106] studied 15 cases of myocarditis after a COVID-19 mRNA vaccine using EMB and immunohistochemical analysis. In nine of these patients, the Spike protein was found in CMs.

Furthermore, vaccinated mice showed signs of myocarditis 2 days after injection of the second dose of BNT162b2 vaccine [81], and free Spike antigen was detected in the blood of adolescents and young adults who developed myocarditis following a COVID-19 mRNA vaccine [201].

Finally, we have included in Table 3 immune cells infiltrating the myocardium in three different types of subjects affected by myocarditis.

**Table 3 vaccines-11-00747-t003:** Immune cells infiltrating the myocardium in three different types of subjects affected by myocarditis.

Immune Cells in the Myocardium	EMB in Vaccinated	EMB in SARS-CoV-2 Infection	Autopsy in SARS-CoV-2 Infection
CD68 macrophages	Yes	Yes [226,227,228]	Not reported
CD3 lymphocytes	Yes	Yes [226,227,228]	Not reported
CD4 lymphocytes	No	Not reported	Yes [229,230]
CD8 lymphocytes	No	Not reported	Yes [231]
NET	No	Not reported	Yes [231]

CD (cluster of differentiation); EMB (endomyocardial biopsy) in living subjects. NET (neutrophil extacellular traps).

We are now confident that Spike-specific activated T lymphocytes, macrophages, and Spike protein can reach the myocardium after vaccination, but the “Immune Black Hole” prevents us from knowing any interactive modalities between these, and possibly other, cellular components.

Since the natural history of myocarditis does not end after the immediate period following diagnosis, as it can also evolve silently creating the preliminary conditions that could lead to dangerous arrhythmias and sudden death, we would like to bring you some important elements.

If we use CRM images, we can monitor the LGE pattern over time. In the acute phase, CMR allows to verify if there is inflammation/edema, increased interstitial space, and LGE [232]. LGE on CMR imaging signifies myocardial fibrosis or scars [233]. LGE presence is a strong risk marker in patients with suspected myocarditis [234], and LGE-assessed myocardial fibrosis has been shown to be a predictor for outcome in the same patients [235]. Georgiopoulos and colleagues [218] conducted a meta-analysis and demonstrated that the presence and location of LGE may identify a subgroup of patients with acute myocarditis who warrant more intensive clinical surveillance for adverse cardiac events. Indeed, anteroseptal location but not LGE extent was also associated with the clinical outcome. Finally, LGE in basal and mid lateral segments have a better prognosis than cases with LGE localized to the septal segments [218,236,237]. Indeed, in milder cases of myocarditis, the subepicardial layer, especially in the posterolateral wall, presents LGE, while in the most severe cases LGE can be more diffuse and circumferential [236,237]. LGE is present in many cases of myocarditis following a COVID-19 mRNA vaccine [238,239,240] and is likely a robust prognostic marker in children and adults with myocarditis [218].

Among the patients studied by Kracalik et colleagues [241], a subgroup of 151 patients were investigated with MRI and over 50% presented abnormal results (LGE and/or edema), after 90 days from the onset of myocarditis. Additionally, two patients with LGE also had atrial or ventricular arrhythmias. Although there are few cases of arrhythmia associated with the LGE phenomenon, this data reinforces our concern as it demonstrates that scarring can be arrhythmogenic. Furthermore, LGE is a strong and independent predictor of cardiac mortality in patients with myocarditis [234]. It must always be remembered that in clinical practice, there can be complete healing with restitutio ad integrum (complete restoration of the initial conditions) and healing with scarring results and the two types are not superimposable.

Finally, myocarditis can be a potentially lethal complication following an mRNACOVID-19 vaccination [242], but inflammatory infiltration of the myocardium may be different in autopsy examinations (predominantly composed of lymphocytes CD4), than data provided by the EMB (predominantly composed of macrophage CD68^+^).

Voleti and colleagues [243] determined that the risk of myocarditis is more than seven times higher in people who have been infected with SARS-CoV-2 than in those who have received the vaccine. Acute cardiovascular complications of a COVID-19 infection include myocarditis, pericarditis, acute coronary syndrome, heart failure, pulmonary hypertension, right ventricular dysfunction, and arrhythmia [244]. The risk of incident cardiovascular disease extends well beyond the acute phase of COVID-19 [245]. Long-term follow-ups show an increased incidence of arrhythmia, heart failure, acute coronary syndrome, right ventricular dysfunction, myocardial fibrosis, hypertension, and diabetes mellitus [244]. In a series of autopsies, the presence of SARS-CoV-2 virus was found in the myocardium which does not necessarily cause an inflammatory reaction consistent with clinical myocarditis [246]. Tavazzi et al. [227] described a case of myocarditis in a COVID-19-positive patient, where a direct viral infection was seen by an endomyocardial biopsy and the immune cells present in the myocardium were T memory cells and CD68^+^ macrophages. Weckbach and colleagues [228] found in five patients with COVID-19 and myocarditis a prevalent infiltrate of CD68^+^ macrophages with few CD3^+^ T cells. Despite a short history of disease, they observed fibrosis in all patients. This finding of fibrosis in all five patients examined with EMB should alarm us since there is an evident risk of arrhythmias related to fibrosis.

Finally, as reported in Table 3, the inflammatory infiltrate in the myocarditis of living subjects appears to be different compared to the autoptic data, even if it appears to be practically identical in the cases of myocarditis from COVID-19 where the patient did not die, and in the cases of myocarditis following COVID-19 mRNA vaccines.

A prolonged follow-up including CMR appears appropriate for all cases of myocarditis because many patients are at high risk for life-threatening outcomes.

We should remember that there is a differential risk of myo-pericarditis based on age, where vaccinated male subjects, younger than 40 years of age, have a higher risk than infected subjects [247,248]. Conversely, in older subjects, the risk of myo-pericarditis is higher in infected subjects than in vaccinated subjects [243].

Tuvali and colleagues [249] published a retrospective cohort study of 196,992 adults after a COVID-19 infection and they discovered that the incidence of myocarditis was 0.0046% (46 cases per million), while in the control cohort the incidence was the same (0.0046%). These authors state that they did not observe an increased incidence of either pericarditis or myocarditis in adult patients recovering from a COVID-19 infection. Patone and colleagues [247] showed that, in adults younger than 40 years old, the number of excess myocarditis events per million people was higher after a second dose of mRNA-1273 than after a positive SARS-CoV-2 test (97 cases per million versus 16 cases per million of COVID-19 patients, respectively). Finally, according to the FDA and EMA agencies, the risk of myo-pericarditis following COVID-19 mRNA vaccines is about 100 per million [250].

Multisystem Inflammatory Syndrome in children (MIS-C) and adults (MIS-A) are a late complication of the SARS-CoV-2 infection [187,188,189,190,191]. MIS following COVID-19 mRNA vaccines (MIS-V) is a serious adverse event and there are many pediatric case reports that begin 4–6 weeks after the first vaccine dose [193,194].

SARS-CoV-2 can persist for a longer period in the intestines, particularly in children, and most children with MIS-C present with intestinal symptoms, including pain, vomiting, or loose stools (250). In Spike protein, there is one potential superantigen motif near its S1/S2 cleavage site [251] which can lead to superantigen-mediated T cell activation and proliferation [252]. A potential explanation for the delayed presentation in MIS-C is viral persistence and the repeated stimulation of intestinal myeloid cells and T cells (250). Broad T cell activation in children with MIS-C and autoantibody repertoiressuggest that such T cells can also lead to widespread B cell activation and a loss of tolerance (250). Furthermore, acute MIS-C was characterized by high systemic inflammatory cytokines such as IL-1β, IL-6, IL-8, IL-10, IL-17, and IFN-γ [253].

We always remember that each vaccination determines a strong production and secretion of pro-inflammatory cytokines [37,38]. What happens next also depends on how strong this pro-inflammatory response was. Unfortunately, in many cases of MIS-V the markers of inflammation used are few and often include only C-reactive protein, ferritin, and procalcitonin [254]. We have not been able to find MIS-V studies that test the three main pro-inflammatory cytokines (IL-1β, IL-6 and TNF-α), although we did find a case report in which serum IL-6 values were of 566.0 pg/mL [255], while an IL-6 concentration higher than 37.65 pg/mL was predictive of in-hospital death in patients with a SARS-CoV-2 infection [256]. In 16 cases of MIS-C, levels of 14 of 37 cytokines/chemokines (including IL-6, IL-18, and TNF-α), were significantly higher in children with MIS-C compared to those without, irrespective of age or sex [257].

## 11. Conclusions

We think that a series of adverse events following COVID-19 mRNA vaccines (early systemic reactions, arrhythmias, myo-pericarditis, and multisystem syndromes) can be integrated into a new paradigm that we have called the “Inflammatory Pyramid (IP)” (Figure 1). The different degrees of the IP express the levels of inflammation determined by the vaccination that persists over time due to a prolonged antigenic stimulation by the Spike protein, which determines a prolonged and differentiated production of cytokines associated with an immune and non-immune cellular activity as we have described in this review.

The progression of the inflammation degree over time, starting from the injection date of the first vaccine dose, is related to the severity of the disease, as MIS begins one month after the initial stimulus (infection or vaccination).

While systemic symptoms and arrhythmias can be produced by pro-inflammatory cytokines, in the development of myo-pericarditis the cells participating in inflammation cooperate with pro-inflammatory cytokines. Finally, in MIS it seems that a prolonged antigenic stimulation is necessary to generate a strong secretion of pro-inflammatory cytokines and an energetic cellular response.

## Figures and Tables

**Figure 1 vaccines-11-00747-f001:**
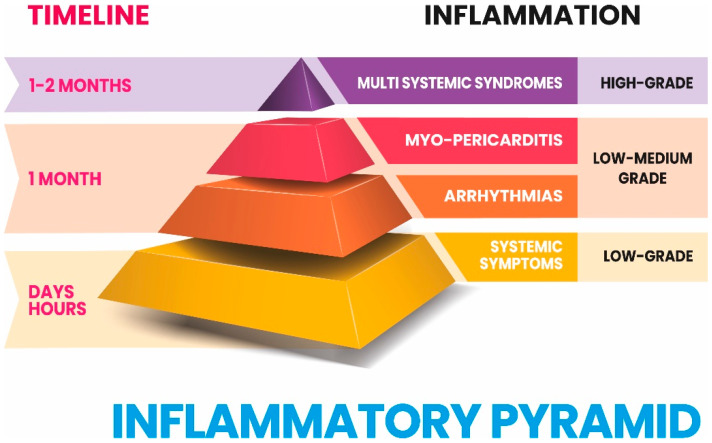
Inflammatory Pyramid (IP) following COVID-19 mRNA vaccines. In our hypothesis, we correlated the timeline of COVID-19 mRNA vaccine adverse events to the degree of inflammation that can occur after their injection. On the left side, we have diversified the time of onset of adverse events, based on the scientific literature presented here. On the right side, we stratified the degree of inflammation into three progressive levels starting from the base of the IP to reach its apex: low-grade, low-medium grade, and high grade of inflammation. The levels of the pyramid are occupied by adverse events, which start from the base and lead to the apex with this increasing degree of clinical complexity: systemic symptoms, heart disease, hyperinflammatory multisystem syndromes (MIS). The timeline correlates with the degree of inflammation and both relate to the different severity of the clinical manifestations temporally associated with the first injection of COVID-19 mRNA vaccines.

**Table 1 vaccines-11-00747-t001:** Kv 4.3, Kv 11.1 or hERG, and Kv 7.1 are voltage-gated potassium channels.

Channel	Current	Current Type	Ion	AP Phase	Interference
K_v_ 4.3	*I* _to_	Outward	K^+^	1	IL-1β, TNF-α
L-type	*I_CaL_*	Inward	Ca^2+^	2	IL-1β, IL-6
K_v_ 11.1 hERG	*I* _Kr_	Outward	K^+^	3	IL-6, TNF-α
K_v_ 7.1	*I* _Ks_	Outward	K^+^	3	TNF-α
Kir 2.1/2.3	*I* _K1_	Inward	K^+^	4	

## Data Availability

All the data can be requested from the corresponding author.

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
