# Peer review of "COVID-19 mRNA Vaccines: The Molecular Basis of Some Adverse Events"

_vaccines, 2023, doi:10.3390/vaccines11040747_

Round 1

Reviewer 1 Report (Previous Reviewer 1)

Everything is fine.

Author Response

Dear reviewer.

We thank you very much for your positive opinion.

Best regards,

Dr Girolamo Giannotta.

March 21, 2023.

Reviewer 2 Report (New Reviewer)

Dear authors, your manuscript is interesting and in my opinion deserves to be published.

Here some comments to improve the quality of your manuscript:

- Introduction "In the post-marketing period, some adverse events (AEs) were temporally associ-30 ated with the injection of COVID-19 mRNA vaccines." Add reference.

-  "The depolarization and repolarization parameters (QRS duration and QT interval) decreased significantly after the vaccine with 143 increasing heart rate." Sentence duplicated at line 142 and 147, please reformulate.

- Lines 154 - 155 "Although it doesn’t seem like it, the monthly increase in cases is statistically significant, practically doubling" How was the statistical significance assessed? Please explain it or delete the sentence. 

- "8.1.6. Young males: the favorite target." About this section, I would be cautious in affirming that adolescent are more stressed than adults, I do not see clear evidence about this. You report "Adolescence is accompanied by increased exposure to stressors", but the comparison is versus children, not versus adults. This doesn't explain the young males being the favorite target. Please reformulate this section.

- Conclusion: this section is too long, please summarize it stating the relevant points. 

For the abovementioned reasons, I recommended a minor revision.

Best regards

Author Response

Dear reviewer.

Thank you for your suggestions which have allowed us to improve our study.

In the attached file there are our answers to your suggestions.

Best regards,

Dr Girolamo Giannotta.

March 21, 2023.

This manuscript is a resubmission of an earlier submission. The following is a list of the peer review reports and author responses from that submission.

Round 1

Reviewer 1 Report

In this paper, the authors did a comprehensive review on a very important subject: COVID 19 mRNA vaccine-induced inflammation, especially myocardial inflammation. The paper is nicely written, and the review is of great importance for the journal's readers. I do not have objections regarding the paper. However, I strongly suggest the authors to consider the following: 1) provide a table highlighting the potential diferences between SARS-CoV-2 induced myocardial inflammation and mRNA vaccine-induced myocardial inflammation. The information has already been provided in the original text, but I think that it would be important to emphasize that; 2) I think that the authors may proceed with a quick re-check of the literature, searching for cases of myocardial inflammation following a first injection with a ChAdOx1 vaccine followed by mRNA vaccine injection. I was able to find at least one case of this circumstance (Mörz M. Vaccines 2022; 10: 1651). This would be also of interesting because this has been a common practice in many countries.

Author Response

Many thanks for your comments and suggestions which have allowed us to improve the quality of our study.

In the attached pdf there are our answers to your comments and suggestions.

Best regards,

Dr Girolamo Giannotta.

February 14, 2023.

Reviewer 2 Report

This is a nice paper written as a review, and using multiple references to build an interesting hypothesis about the "inflammatory pyramid". Although this is a hypothesis the authors use extensively references with experimental and human data, thus  it is a plausible hypothesis. Overall, this is a useful and interesting review.

Author Response

Many thanks for your positive evaluation of our paper.

Best regards,

Dr Girolamo Giannotta.

February 14, 2023.

Reviewer 3 Report

This is huge effort to read/summarize 239 articles for a manuscript, however author prefer to select articles with the same direction of their hypothesis. This is review article, authors need to harmonize positive and negative results related with the vaccine and should summarize all results for potential readers (potential readers will agree your hypothesis or not).

In this article authors presented their opinions about the inflammation and hyperinflammation related with mRNA vaccines. Firstly, and also very important point, “vaccinology/epidemiology- training/education/experience” is mandatory in the preparation of this kind of articles (efficacy, side effects and epidemiological consequences of vaccines). Abstract of this article start with the phrase like this “Each injection of any known vaccine results in a strong expression of pro-inflammatory cytokines”. If you prefer to use this information, you should start in the Introduction section and need to add all references related with this topic for each vaccine. Second sentence of the Abstract is “COVID-19 mRNA vaccines would not escape this rule” is not a phrase for scientific article. Instead of writing a review by interpreting the articles published in the literature in line with Author’s hypotheses, it is necessary to conduct experimental or field studies that support these hypotheses. I think it would be firstly reviewed their data on the inflammation/hyperinflammation vaccine relationship in experimental models, taking the scientists who are experienced in this field with them.

1-    What is the best definition for low grade inflammation or high-grade inflammation for your hypothesis? Which inflammation markers? CRP, PCT, ferritin? High grade inflammation is equal to MIS (C/N/A or V)? Do you have any information about the comparison of inflammation parameters between early and late period of the disease (or in long-COVID patients) or early-late period after vaccination. “Authors concludes that clinical manifestations are correlated to the increasing degree of inflammation: symptoms, heart disease and syndromes” however they need to prove it. If not, this is hypothetical approach and you need to show your data first (about inflammation course after disease or immunization).

2-    I don’t understand why authors add “The cardiac Action Potential (AP)” and other basic cardiac physiology (article is too long, and difficult to follow)

3-    Authors mentioned that “Increased risk of myocarditis / pericarditis has been as-141 sociated with the second dose of BNT162b2 and both doses of mRNA-1273 [67]. In one 142 study the highest risks were observed in males of 12 to 39 years [67], while Wong and 143 colleagues [68], demonstrated that an increased risk of myocarditis or pericarditis was 144 observed after COVID-19 mRNA vaccination and was highest in men aged 18–25 years 145 after a second dose of the vaccine.”

Before pandemic, myocarditis due to other infectious causes are higher in boys/men, especially 16-25 years. The gender difference for myocarditis for this age group is classical findings for infectious myocarditis. Could author explain why myocarditis is higher in men aged 16-25 years old after vaccination? Authors prefer to use “Young males: the favorite target” for their hypothesis however they did not discuss the myocarditis cases in these age group before pandemic. Like other part of the study, author prefer to use all possible explanation (related/unrelated) to support their hypothesis. If you plan to discuss vaccine related side effects like myocarditis/pericarditis, you also need to evaluate the effect of disease itself on this condition.

4-    In the Abstract; author mentioned that “We HYPOTHESIZE that we can move from a limited pro-inflammatory condition to conditions of increasing expression of pro-inflammatory cytokines 16 that can culminate in multisystem hyperinflammatory syndromes following COVID-19 mRNA 17 vaccines (MIS-V)”. In the text, they cited some studies about MIS-C. MIS-C is an important and serious conditions affecting children. There are some possible explanations for MIS-C, superantigen motif, genetic background or microbiota composition. However, it should be noted that there are some epidemiological changes for MIS-C regarding the circulating variants. Comparing the first two years of pandemic, during the last year of pandemic (after Omicron variant), incidence of MIS-C decreased/disappeared. Another important point for pediatric age group, MIS-C incidence were lower in children who were vaccinated than the unvaccinated ones.

5-    Part 8.1.1 starts “In several studies, myocarditis / pericarditis are more frequent after injection of the second dose of Pfizer / Biontech mRNA vaccine; while myocarditis / pericarditis can occur both after the first and second injection of Moderna’s mRNA vaccine. Considering the time interval between the two vaccine doses; one would think that it takes less than 28 days to produce this AEs (about 23-27 days for the BNT162b2 vaccine)”. Authors need to add reference for this kind of important information. Also need to harmonize to use BNT162b2 vaccine instead of Pfizer/BionTech vaccine, also need to use original vaccine name for Moderna vaccine, too.

6-    Authors prefer to use all references related with the vaccine side effects, however they need to evaluate all these references carefully.

Example: “Li and colleagues [75] conducted a study in a Balb / c mouse model and found that intravenous injection of a COVID-19 mRNA vaccine rapidly resulted in multifocal myopericarditis, while intramuscular injection produced signs 7 days after injection. (myocardial edema, and occasional foci of cardiomyocyte degeneration). However, these histopathological changes worsened after the second dose. Since the injected vaccine was BNT162b2, this study may indicate that two shots are needed with this vaccine to produce myopericarditis, as occurs predominantly in humans.”

In this in vivo study (Li and colleagues) showed only the IV group showed histopathological changes of myopericarditis, not the group that received the intramuscular BNT162b2 mRNA COVID-19 vaccine. However, it is important to note that the authors administered 0.25 µg of BNT162b2 per gram of body weight to the mice, which is a much larger dose per body weight than would be used in humans (Ref:

Addressing the Elephant in the Room: Intravenous Injection of Coronavirus Disease 2019 mRNA, https://www.ncbi.nlm.nih.gov/pmc/articles/PMC9383553/)

7-    Another example “while Wong and 143 colleagues [68], demonstrated that an increased risk of myocarditis or pericarditis was observed after COVID-19 mRNA vaccination and was highest in men aged 18–25 years after a second dose of the vaccine”. This article published in Lancet and it is very important for all HCPs. However interpretation of the Lancet article author’s like this “Interpretation: An increased risk of myocarditis or pericarditis was observed after COVID-19 mRNA vaccination and was highest in men aged 18-25 years after a second dose of the vaccine. However, the incidence was rare. These results do not indicate a statistically significant risk difference between mRNA-1273 and BNT162b2, but it should not be ruled out that a difference might exist. Our study results, along with the benefit-risk profile, continue to support vaccination using either of the two mRNA vaccines.”

8-    In Table authors showed “Summary of some effects determined by the injection of COVID-19 mRNA vaccines”. In second line they showed“Actions of the Spike protein on the Angiotensin II / AT1 axis” and cited references from 127-138. These references are not related with COVID19 mRNA vaccines. Some of these references are not related with COVID-19 anf published couple of years ago (before pandemic). Need to focus this table, and need to include only data related with mRNA vaccines or LNP, if available.

Final note: An increased risk of myocarditis or pericarditis was observed after COVID-19 mRNA vaccination and was highest in men aged 18-25 years after a second dose of the vaccine. Authors could summarize all data abaout this situation and their concerns about immunization. However, in this article, authors did not evaluate the direct effect of COVID-19 infection (not vaccine) on cardiovascular health. Also they did not prefer to discuss publication about the potential benefits of widespread COVID-19 immunization on disease severity and also mortality. If they prefer to highlight their inflammation hypothesis, they need to start with SARS-CoV-2 related excessive inflammation during the first year of pandemic (before vaccine).  This is huge effort to read/summarize 239 article for a manuscript (inlcude very important article of Yonker et al in 2023, special congratulations fort his updated information), however author need to harmonize positive and negative results related with the vaccine for all available data.

Author Response

(The authors gave the same response as above.)

Round 2

Reviewer 3 Report

Although you try to answer (gently and with reference) the questions and criticisms I have asked, the article is still based on your own view in only one direction. I think that -still- not correct to discuss the issue only over the vaccine, without talking about the short and long-term cardiac effects of the COVID-19 infection.

Author Response

Comments and Suggestions for Authors
Although you try to answer (gently and with reference) the questions and criticisms I have asked, the article is still based on your own view in only one direction. I think that -stillnot correct to discuss the issue only over the vaccine, without talking about the short and long-term cardiac effects of the COVID-19 infection.

Response to the referee comment.
Voleti and colleagues [1] determined that the risk of myocarditis is more than seven times higher in peoplewho have been infected with SARS-CoV-2 than in those who have received the vaccine. Acutecardiovascular complications of COVID-19 infection include myocarditis, pericarditis, acute coronary syndrome, heart failure, pulmonary hypertension, right ventricular dysfunction, and arrhythmia [2]. The risk of incident cardiovascular disease extends well beyond the acute phase of COVID-19 [3]. Long-term follow-up shows increased incidence of arrhythmia, heart failure, acute coronary syndrome, right ventricular dysfunction, myocardial fibrosis, hypertension, and diabetes mellitus [2]. In a series of autopsies, the presence of SARS-CoV-2 virus was found in the myocardium which does not necessarily cause an inflammatory reaction consistent with clinical myocarditis [4]. Tavazzi et al [5] described a case of
myocarditis in a COVID-19–positive patient, where direct viral infection was seen by endomyocardial biopsy and the immune cells present in the myocardium were T memory cells and CD68-macrophages. Weckbach and colleagues [6] found in 5 patients with Covid-19 and myocarditis a prevalent infiltrate of
CD68- macrophages with few CD3+ T cells. Despite a short history of disease, they observed fibrosis in all patients. This finding of fibrosis in all 5 patients examined with EMB should alarm us since there is an evident risk of arrhythmias related to fibrosis.

Finally, as reported in table 3, the inflammatory infiltrate in the myocarditis of living subjects appears to be different compared to the autoptic data, even if it appears to be practically identical in the cases of myocarditis from COVID-19 who did not die, and in the cases of myocarditis following COVID-19 mRNA
vaccines. Prolonged follow-up including CMR appears appropriate for all cases of myocarditis because many patients are at high risk for life-threatening outcomes.

References
1- Voleti N, Reddy SP, Ssentongo P. Myocarditis in SARS-CoV-2 infection vs. COVID-19 vaccination: A systematic review and meta-analysis. Front Cardiovasc Med. 2022 Aug 29;9:951314. doi: 10.3389/fcvm.2022.951314. PMID:36105535; PMCID: PMC9467278.

2- Tobler DL, Pruzansky AJ, Naderi S, Ambrosy AP, Slade JJ. Long-Term Cardiovascular Effects of COVID-19: Emerging Data Relevant to the Cardiovascular Clinician. Curr Atheroscler Rep. 2022;24(7):563-570.
doi:10.1007/s11883-022-01032-8.

3- Xie, Y., Xu, E., Bowe, B. et al. Long-term cardiovascular outcomes of COVID-19. Nat Med 28, 583–590 (2022). https://doi.org/10.1038/s41591-022-01689-3. 

4- Lindner D, Fitzek A, Bräuninger H, et al. Association of Cardiac Infection With SARS-CoV-2 in Confirmed COVID-19 Autopsy Cases. JAMA Cardiol. 2020;5(11):1281-1285. doi:10.1001/jamacardio.2020.3551.

5- Tavazzi G, Pellegrini C, Maurelli M, Belliato M, Sciutti F, Bottazzi A, Sepe PA, Resasco T, Camporotondo R, Bruno R, Baldanti F, Paolucci S, Pelenghi S, Iotti GA, Mojoli F, Arbustini E. Myocardial localization of coronavirus in COVID-19 cardiogenic shock. Eur J Heart Fail. 2020 May;22(5):911-915. doi: 10.1002/ejhf.1828. Epub 2020 Apr 11. PMID: 32275347; PMCID: PMC7262276.

6- Weckbach LT, Curta A, Bieber S, Kraechan A, Brado J, Hellmuth JC, Muenchhoff M, Scherer C, Schroeder I, Irlbeck M, Maurus S, Ricke J, Klingel K, Kääb S, Orban M, Massberg S, Hausleiter J, Grabmaier U. Myocardial Inflammation and Dysfunction in COVID-19-Associated Myocardial Injury. Circ Cardiovasc Imaging. 2021 Jan;14(1):e012220. doi: 10.1161/CIRCIMAGING.120.011713. Epub 2021 Jan 19. PMID: 33463366.

We have inserted these sentences into the manuscript on page 15.
Thank you for your suggestions which have allowed us to improve our study.
Best regards,
Dr Girolamo Giannotta.
February 16, 2023.